# Effectiveness of Booster Doses of the SARS-CoV-2 Inactivated Vaccine KCONVAC against the Mutant Strains

**DOI:** 10.3390/v14092016

**Published:** 2022-09-12

**Authors:** Chanchan Xiao, Jun Su, Chanjuan Zhang, Boya Huang, Lipeng Mao, Zhiyao Ren, Weibin Bai, Huayu Li, Guomin Lei, Jingshan Zheng, Guobing Chen, Xiaofeng Liang, Congling Qiu

**Affiliations:** 1Department of Microbiology and Immunology, Institute of Geriatric Immunology, School of Medicine, Jinan University, Guangzhou 510630, China; 2Affiliated Huaqiao Hospital, Jinan University, Guangzhou 510630, China; 3Department of Pathophysiology, Key Laboratory of State Administration of Traditional Chinese Medicine of the People’s Republic of China, School of Medicine, Jinan University, Guangzhou 510630, China; 4Department of Systems Biomedical Sciences, School of Medicine, Jinan University, Guangzhou 510630, China; 5Department of Food Science and Engineering, Institute of Food Safety and Nutrition, Guangdong Engineering Technology Center of Food Safety Molecular Rapid Detection, Jinan University, Guangzhou 510630, China; 6Shenzhen Kangtai Biological Products Co., Ltd., Shenzhen 518000, China; 7Jinan University-BioKangtai Vaccine Institute, School of Medicine, Jinan University, Guangzhou 510630, China

**Keywords:** SARS-CoV-2, inactivated vaccine KCONVAC, cellular immune responses, variants

## Abstract

As the COVID-19 epidemic progresses with the emergence of different SARS-CoV-2 variants, it is important to know the effectiveness of inactivated SARS-CoV-2 vaccines against the variants. To maximize efficiency, a third boost injection of the high-dose SARS-CoV-2 inactivated vaccine KCONVAC was selected for investigation. In addition to the ancestral strain, KCONVAC boost vaccination induced neutralizing antibodies and antigen-specific CD8 T cells to recognize several variants, including B.1.617.2 (Delta), B.1.1.529 (Omicron), B.1.1.7 (Alpha), B.1.351 (Beta), P.3, B.1.526.1 (Lota), B.1.526.2, B.1.618, and B.1.617.3. Both humoral and cellular immunity against variants were lower than those of ancestral variants but continued to increase from day 0 to day 7 to day 50 after boost vaccination. Fifty days post-boost, the KCONVAC-vaccinated CD8 T-cell level reached 1.23-, 2.59-, 2.53-, and 1.01-fold that of convalescents against ancestral, Delta, Omicron and other SARS-CoV-2 variants, respectively. Our data demonstrate the importance of KCONVAC boosters to broaden both humoral and cellular immune responses against SARS-CoV-2 variants.

## 1. Introduction

Since the World Health Organization announced the global pandemic of coronavirus disease 2019 (COVID-19), vaccination has been the most efficient and cost-effective intervention to control COVID-19. As of July 2022, the global SARS-CoV-2 vaccine injection count has reached more than 12 billion doses, and China has reported a total of 3.4 billion vaccine doses with a vaccination rate of over 88% [1]. Inactivated SARS-CoV-2 virus vaccines have been widely applied in China and many other countries. Of note, the inactivated SARS-CoV-2 vaccine at a high dosage (KCONVAC 5 mg) showed a significantly high degree of humoral immunogenicity, safety, and protective efficacy in a phase I/II clinical trial [2]. As the epidemic progresses with the emergence of different SARS-CoV-2 variants, we are interested in determining the protective ability of KCONVAC against mutant strains.

Currently, the efficiency of vaccines is evaluated by the reduced infection rate and neutralizing antibody production in vaccinated subjects. Several studies have demonstrated that the BNT162b2 and m-1273 mRNA vaccine can activate humoral immunity against the major SARS-CoV-2 mutants that are currently in circulation [3,4,5]. The neutralizing antibody responses to different types of vaccines varied, whereby the response to the inactivated-virus vaccine was similar to the convalescent sera in five epidemic SARS-CoV-2 variants in the UK [6]. However, the cellular immunity against SARS-CoV-2 variants after inactivated vaccination is still unclear.

Cellular immunity plays a decisive role in resisting SARS-CoV-2 infection, especially in the severe evolution after infection. The cellular immune memory effect is also much longer than that of humoral antibodies [7]. Early in the bronchoalveolar lavage fluid of patients with COVID-19, a large amount of CD8+ T-cell infiltration was found, and the induction of the CD8+ T-cell immune response was also related to the severity of COVID-19 [8]. Moreover, in a study of SARS, the immune activity of SARS-specific CD8 T cells was detectable after 11 years [9], indicating the important role of the CD8 T-cell immune response in antiviral immune defense.

Here, based on the previously identified SARS-CoV-2 ancestral and mutant CD8+ T-cell epitopes, we detected the effect of the inactivated vaccine KCONVAC after booster injection with neutralizing antibodies and epitope-specific CD8+ T-cell levels in vaccinated donors and evaluated the efficiency compared with that in convalescent donors.

## 2. Material and Methods

### 2.1. Human Subject Enrollment

This study was approved by the institutional ethics committee of the Guangdong Provincial Centers for Disease Control and Prevention (GZCDC-ECHR-2022P0001).

We recruited a cohort of 43 SARS-CoV-2 vaccinees and 20 convalescent COVID-19 subjects, among which 19 and 6 were HLA-A2 positive, respectively. Nineteen healthy HLA-A2 volunteers with no history of coronavirus disease 2019 (COVID-19) infection were enrolled and included in this study (Appendix A). The volunteers received 5 mg of the inactivated SARS-CoV-2 vaccine KCONVAC (China Shenzhen Lot No. 2021080109) between March and June 2022. None of the participants experienced serious adverse effects after vaccination. Blood samples were collected at baseline (prior to the third vaccination dose) and 7 and 50 days after the third vaccination dose. For convalescent subjects, blood was collected at 7 and 50 days after the last negative report for the SARS-CoV-2 RT–PCR assay (Appendix A). All the sample information was blinded during all the experiments.

### 2.2. Isolation of Plasma and PBMCs

Whole blood was collected in heparinized blood vacutainers and kept on gentle agitation until processing. Plasma was collected after centrifugation. PBMCs were isolated by density gradient centrifugation using lymphocyte separation medium (GE, Chicago, IL, USA). The percent viability was estimated using standard Trypan blue staining. The PBMCs were cryopreserved in fetal bovine serum (LONSERA, Uruguay) with 10% DMSO (Sigma-Aldrich, Burlington, VT, USA) and stored in liquid nitrogen until use.

### 2.3. Neutralizing Antibody Measurement

Diluted serum and equal volumes of 2CCID_50_/µL of different SARS-CoV-2 strains (ancestral; Delta B.1.617.2; Omicron B.1.1.529) were incubated at 37 °C in a 5% CO_2_ incubator for 2 h and then cocultured with Vero E6 cells. After 4 to 6 days of incubation, the neutralizing antibody titers of serum against different SARS-CoV-2 strains were determined according to cytopathic conditions. The above experiments were performed in the P3 biosafety laboratory.

### 2.4. HLA-A2 Restricted T-Cell Epitope Selection

The spike (S), membrane (M), nucleocapsid (N), and ORF protein sequences of the SARS-CoV-2 ancestral-Hu-1 strain (NC_045512.2) were used for T-cell epitope prediction with the “MHC I Binding” tool (http://tools.iedb.org/mhci, 1 February 2022). The prediction method used was IEDB Recommended 2.22 (NetMHCpan EL) with the MHC allele selected as HLA-A*02:01, the most frequent class I HLA genotype among the Chinese population [10,11]. All predicted epitopes containing the same amino acid residue corresponding to the mutation from B.1.1.7 (Alpha), B.1.351 (Beta), P.3, B.1.526.1 (Lota), B.1.526.2, B.1.618, B.1.617.2 (Delta), B.1.617.3, and B.1.1.529 (Omicron) were compared. The peptide with the best prediction score was used as the candidate epitope for the ancestral strain. Epitopes from mutant strains were excluded with peptide length >12 aa and predicted antigen presentation ability by VaxiJen 2.0 (http://www.ddg-pharmfac.net/vaxijen/VaxiJen/VaxiJen.html, 1 February 2022). All the epitopes were validated in our previous studies [12,13].

### 2.5. Generation of Antigen-Specific HLA-A2 Tetramer

Thirty microliters of peptide-exchanged monomer (BioLegend, San Diego, CA, USA) formed in the above steps was mixed with 3.3 µL of PE streptavidin (BioLegend, San Diego, CA, USA) on a new plate and incubated on ice in the dark for 30 min. Then, 2.4 µL of blocking solution (1.6 µL of 50 mM biotin) (Thermo Fisher, Waltham, MA, US) plus 198.4 µL of PBS was added to stop the reaction and it was incubated at 4–8 °C overnight.

### 2.6. Cell-Surface Antibodies and Tetramer Staining

PBMCs were isolated from the peripheral venous blood of healthy donors and SARS-CoV-2 vaccinees. The HLA-A2+ donors were identified by using flow cytometry without subtype identification. Briefly, 10^6^ PBMCs were stained with PE-conjugated anti-human HLA-A2 antibodies (BioLegend, San Diego, CA, USA) at 4 °C in the dark for 30 min and acquired by using a FACS Canto flow cytometer (BD). HLA-A2-positive PBMC samples were further stained with PE-labeled tetramers (homemade) plus APC-labeled human CD8 antibodies (BioLegend, San Diego, CA, USA).

### 2.7. Antigen-Specific CD8 T-Cell Stimulation and Function Detection In Vitro

With the previously reported artificial antigen-presenting cell system from our studies and others [14], HLA-A2-expressing T2 cells were loaded with peptides for subsequent CD8 T-cell activation. Briefly, T2 cells were treated with 20 µg/mL mitomycin C for 30 min to stop cell proliferation and loaded with the indicated epitope peptides for 4 h. CD8 T cells were purified from PBMCs with EasySep Human negative selection (Stemcell, Vancouver, BC, Canada) with a purity over 95%. CD8 T cells (0.25 × 10^6^) isolated from vaccinated donors were cocultured with 0.25 × 10^6^ peptide-loaded (ORF1a 1707-16, ORF1a 2225-34, ORF1a 2230-38, S 2-11, M 82-90, ORF1a 2340-49, and ORF1a 3683-92) T2 cells stained with 5 µmol/L CFSE (TargetMol) and cocultured with 1 µg/mL anti-human CD28 antibodies (BioLegend, San Diego, CA, USA) and 50 IU/mL IL-2 (SL PHARM, Recombinant Human Interleukin-2(125Ala) Injection). Then, the 50 IU/mL IL-2 and 20 µM mixed peptides were supplemented every two days. The T-cell activation marker CD69 (BioLegend, San Diego, CA, USA) was evaluated after 16 h, while tetramer-specific CD8 T cells and the apoptosis marker Annexin V-APC (BioLegend, San Diego, CA, USA) on T2 cells were evaluated after 7 days. On day 7, the cells were restimulated with peptides for 4 h in the presence of Leuko Act Cktl with GolgiPlug (BD, New York, NY, USA) plus 50 IU/mL IL-2, and the production of IFN-γ and granzyme B was checked with PerCP-conjugated anti-human IFN-γ (BioLegend, San Diego, CA, USA) and FITC-conjugated anti-human granzyme B (BioLegend, San Diego, CA, USA) was stained and acquired with a FACS Canto flow cytometer (BD).

### 2.8. ELISpot Assays

Thawed PBMCs were rested for 3–4 h at 37 °C in RPMI 1640 media supplemented with 10% fetal bovine serum (LONSERA, Uruguay). Cells were then stimulated with a 20 µM peptide pool corresponding to ancestral/mutant peptides from B.1.617.2 (Delta) and B.1.1.529 (Omicron). Cell suspensions were transferred to precoated human IFN-γ ELISpot Plus kits (MabTech, Stockholm, Sweden) and developed after 3–4 days according to the manufacturer’s instructions. Spots were imaged and counted using an ELISpot reader (Mabtech).

### 2.9. Statistical Analysis

The data were analyzed by one-way ANOVA and paired-samples t tests for statistical significance by using GraphPad Prism 8 (GraphPad Software Inc., San Diego, CA, USA) and SPSS 22.0 software (SPSS Inc., Stanford, CT, USA). A *p* value less than 0.05 was considered to be statistically significant.

## 3. Results

### 3.1. Neutralizing Antibodies against the Ancestral, B.1.617.2 (Delta), and B.1.1.529 (Omicron) Inactivated KCONVAC Vaccines

To evaluate the adaptive immune response after the third dose of the KCONVAC vaccine, blood was collected from the 43 healthy donors (Appendix A) at three time points: before (baseline), 7 days after injection, and 50 days after injection (Figure 1a). We first measured the neutralizing antibodies against ancestral, Delta (B.1.617.2), and Omicron (B.1.1.529) strains. As expected, the neutralizing antibody titers increased sequentially against the ancestral, Delta, and Omicron strains (Figure 1b). Since the inactivated vaccine KCONVAC was manufactured using an ancestral strain, it is unsurprising that a detectable level of neutralizing antibodies against ancestral, but not Delta and Omicron strains, existed in the samples prior to the boost vaccination (Figure 1b). At the early time (day 7) after the third dose injection, the neutralizing antibody titer increased 12.24-, 3.01-, and 3.07-fold against the ancestral, Delta, and Omicron strains, respectively (Figure 1b). The absolute neutralizing antibody titer against the Delta strain was higher than that against the Omicron strain (Figure 1b). At 50 days post-injection, the neutralizing antibody titer continued to increase 2.21-, 1.79-, and 1.32-fold (day 50/day 7) against the ancestral, Delta, and Omicron strains, respectively (Figure 1c). All these results indicate that even lower titers and boost injection of the KCONVAC vaccine could induce neutralizing antibodies against SARS-CoV-2 Delta and Omicron variants.

### 3.2. Antigen-Specific CD8 T Cells against Ancestral, Delta, and Omicron for Inactivated KCONVAC Vaccine

We then checked cellular immunity after the KCONVAC boost vaccination. Based on the previously identified HLA-A2 positive epitopes, we prepared tetramers recognizing ancestral and other variant strains (Table 1). With these epitope-specific tetramers, we measured SARS-CoV-2-specific CD8 T-cell responses after vaccination (Figure 2a). As expected, two epitope (V1 and V2)-specific CD8 T cells increased sequentially after vaccination (Figure 2b,c). With the mutant V1 and V2 tetramers, which were prepared with specific epitopes on the Delta and Omicron strains, respectively, we could detect how CD8 T cells recognized the corresponding variant strain [13]. As shown in Figure 2b,c, both mutant V1 and V2 tetramer-specific CD8 T cells were detected in the KCONVAC boost-vaccinated subjects. Both mutant V1- and V2-specific CD8 T cells were lower than those of ancestral CD8 T cells, with the lowest percentage in the mutant V2 group (Figure 2d). However, both mutant V1- and V2-specific CD8 T cells indicated further expansion compared with the ancestral group from day 7 to day 50 post vaccination (Figure 2e). All these results indicated that KCONVAC vaccination-induced CD8 T cells could partially recognize Delta and Omicron variants, which suggested that the KCONVAC vaccine could protect the subjects against SACS-CoV-2 infection by Delta and Omicron variants to some degree.

### 3.3. Antigen-Specific CD8 T Cells against Other Variants for Inactivated KCONVAC Vaccine

We further measured CD8 T cells specific to other SARS-CoV-2 variants, including B.1.1.7 (Alpha), B.1.351 (Beta), P.3, B.1.526.1 (Lota), B.1.526.2, B.1.618, and B.1.617.3, using the V3, V4, V5, V6, V7, V8, and V9 tetramers, respectively (Table 1).

All seven epitope-specific CD8 T cells exhibited a sequential increase after boost vaccination with KCONVAC (Figure 3a,b). Similar to the responses to the Delta and Omicron strains, the KCONVAC-immunized CD8 T cells could recognize all the mutated epitopes caused by these variants, except for the Beta (B.1.351) variant (Figure 3a–c). All the mutant epitope-specific CD8 T cells were also lower than the ancestral CD8 T cells (Figure 3c). However, the mutant epitope-specific CD8 T cells exhibited expansion after vaccination (Figure 3a,b) and even more expansion than ancestral CD8 T cells from day 7 to day 50 after vaccination (Figure 3d). These data indicate that KCONVAC vaccination-induced CD8 T cells could partially recognize the Alpha, P.3, Lota, B.1.526.2, B.1.618, and B.1.617.3 variants, which suggested that the KCONVAC vaccine could protect the subjects against SACS-CoV-2 infection of these variants to some degree.

### 3.4. Cytotoxic Function of Antigen-Specific CD8 T Cells Induced by Inactivated KCONVAC Vaccine

To check the cytotoxic function of the antigen-specific CD8 T cells induced by the inactivated KCONVAC vaccine, we stimulated CD8 T cells with artificial antigen-presenting T2 cells loaded with mixed epitopes. As shown in Figure 4a, the T-cell activation marker CD69 significantly increased after epitope stimulation in vitro. We labeled the target cell (epitope-loaded T2 cells) with CSFE. The CFSE+ Annexin V+ apoptosis target cells increased in the samples containing the vaccinated CD8 T cells from day 0 to day 50 after KCONVAC vaccination (Figure 4b). The secretion of both IFN-g (Figure 4c) and GZMB (Figure 4d) increased significantly from day 0 to day 50 after vaccination. To further validate this hypothesis, we also analyzed the T-cell responses of the KCONVAC vaccine participants by ELISpot. The results showed the same cytotoxic function tendency for KCONVAC vaccine-induced CD8 T cells (Figure 4e,f). Our data showed an overall increase in CD8 T-cell cytotoxic function after the KCONVAC booster vaccination.

### 3.5. Comparison of CD8 T-Cell Responses against Delta and Omicron Strains between KCONVAC Boost Vaccination and Natural Infection

It has been clear that a high-dose boost KCONVAC vaccine could induce antigen-specific CD8 T-cell responses against both ancestral and several variant strains. However, it was unclear whether the degree of antigen-specific CD8 T cells was sufficient to protect the host. To answer this question, we planned to compare the antigen-specific CD8 T-cell responses between KCONVAC-vaccinated and naturally infected subjects. We first detected antigen-specific CD8 T cells at the same time points in convalescent subjects who were infected with SARS-CoV-2 Delta strains in May 2021 (Appendix A). The Delta- and Omicron-specific CD8 T cells were first measured using the same tetramers (Figure 5a). Unlike the slight decrease in antigen-specific CD8 T cells from day 7 to day 50 in convalescents, the KCONVAC vaccination induced a continuous increase in antigen-specific CD8 T cells (Figure 5b,c) and an even higher percentage of antigen-specific CD8 T cells than convalescents at day 50 (after boosted vaccination of infection) (Figure 5b,c). For the variants at day 50, the vaccinated subjects showed similar levels of CD8 T cells in the Delta group (Figure 5b) but low levels in the Omicron group (Figure 5c) when compared with the convalescents. Taken together, these results suggest that KCONVAC boost vaccination could induce antigen-specific CD8 T-cell levels similar to those of natural infection, which suggests sufficient protection against SARS-CoV-2 ancestral, Delta, and, to some degree, Omicron strains.

### 3.6. Comparison of CD8 T-Cell Responses against Other Variants between KCONVAC Boost Vaccination and Natural Infection

We further compared CD8 T-cell responses against other SARS-CoV-2 variants between KCONVAC boost vaccinees and convalescents. We first checked the recognition degree of CD8 T cells against B.1.1.7 (Alpha), B.1.351 (Beta), P.3, B.1.526.1 (Lota), B.1.526.2, B.1.618, and B.1.617.3 in the convalescents using the V3, V4, V5, V6, V7, V8, and V9 tetramers, respectively (Figure 6a,b). Similar to the scenario of Delta and Omicron, KCONVAC-vaccinated CD8 T cells only recognized these variants partially, approximately 85.21% at day 50 compared with convalescents (Figure 6c). This indicates that the KCONVAC boost-vaccinated CD8 T cells could recognize most of the SARS-CoV-2 variants.

## 4. Discussion

With the development of the COVID-19 pandemic, the increased transmission of SARS-CoV-2 mutants, especially Delta and Omicron stains, has caused a considerable social and economic burden. On 25 December 2021, the World Health Organization listed the Omicron variant as a variant of concern. Compared with other SARS-CoV-2 variants, the Omicron variant has a large number of mutation sites on the Spike protein, which leads to stronger immune evasion and affects the antibody neutralizing ability in vaccinated populations [15]. However, vaccine protection is dependent not only on antibody-dependent neutralizing capacity, but also on cellular immunity and immune memory. Therefore, this study evaluated the protection of high-dose inactivated virus vaccines to the current mainstream mutants after boosting specific antibodies and cellular immunity.

The KCONVAC Phase I/II clinical trial data showed that neutralizing antibodies were 2.7 times higher in the vaccinees than those of recovered patients, the highest among inactivated virus vaccines [16,17]. For the neutralizing antibody analysis, Vero-E6 cells (African green monkey kidney cells) infected with Delta or Omicron stains were used as the target cells cultured in the serum of vaccine recipients, and the neutralization ability of the serum was determined by measuring the virus titer in the P3 biosafety laboratory [18]. Our results showed that the neutralizing antibody levels of Delta and Omicron mutants after booster immunization with high-dose inactivated virus vaccine gradually increased with the prolongation of immunization time but were still lower than those of the ancestral. These results are consistent with current reports of decreased neutralizing capacity of antibodies against Delta and Omicron mutants following immunization with the COVID-19 vaccine [19,20]. The binding ability of neutralizing antibodies in serum to the Delta variant and Omicron variant RBD after two doses of the inactivated vaccine produced by Sinopharm WIBP was measured by ELISA, and the results showed that the Omicron variant could escape the antibodies induced by the ancestral strain and the inactivated vaccine. The binding capacity of its RBD was significantly reduced, much more than that of the Delta variant [19]. In an evaluation of humoral immunity with a heterologous BNT162b2 mRNA vaccine booster in participants who received a two-dose regimen of CoronaVac, an inactivated vaccine used globally, followed by a BNT162b2 booster after two doses of a heterologous CoronaVac prime vaccine, elevated levels of virus-specific antibodies were induced, and although Omicron neutralization was undetectable in participants receiving the two-dose CoronaVac regimen, the BNT162b2 booster increased Omicron-neutralizing activity 1.4 times compared to the two-dose mRNA vaccine.

Nevertheless, virus-specific T cells were still detected in vaccinated subjects with neutralizing antibodies testing negative for SARS-CoV-2, which does not mean they are not immune to COVID-19. According to the study of Marcus Buggert et al., specific T cells can still be detected in vaccinated populations with negative neutralizing antibodies and have the cytotoxicity of target cells [21].

In the study of Dallmeier, K et al., the effectiveness of existing COVID-19 vaccines remained high in preventing severe illness and death [22]. ChAdOx1 nCoV-19 or BNT162b2 vaccines provided limited protection against symptomatic disease caused by the Omicron variant after primary immunization and significantly increased protection after booster immunizations, but this protection diminished over time [23]. This further indicated that although neutralizing antibodies declined, cellular immunity and immune memory might play a longer-term protective role.

An artificial antigen-presenting cell (aAPC) system was used in this study, providing a convenient protocol to assess the degree of CD8+ T-cell activation with no need for a high-level biosafety laboratory. Ancestral and mutant CD8+ T-cell-specific epitopes were loaded onto HLA-A2 molecules on the surface of T2 cells as target cells and cocultured with CD8+ T cells isolated from the vaccinated population. Then, cytotoxicity and the T-cell-activated markers CD69, IFN-γ and GZMB were analyzed. Due to the limitation of the T2 cell model and the restricted HLA-A2 subtype, this study only focused on HLA-A2-positive populations. First, there was no significant difference in the proportion of epitope-specific CD8+ T cells between the ancestral strains or the Delta and Omicron strains before booster immunization. The difference appeared on day 7 and gradually expanded by day 50. This phenomenon was more pronounced in Omicron mutants than in Delta mutants. Next, among the currently emerging SARS-CoV-2 mutant strains, there was no significant difference in the proportion of epitope-specific CD8+ T cells before booster immunization. However, the increased proportion of ancestral epitope-specific CD8+ T cells over mutants resulted in a progressively widening difference. This finding indicates that the CD8+ T cells specific for the ancestral or the Delta and Omicron epitopes increased to different degrees after booster immunization. Last, with the stimulation of mixed polypeptides (ancestral and mutant), cellular immunity was enhanced after boosting immunization, which was manifested as the increased expression of the cytokines IFN-γ and GZMB and the enhanced killing ability of target cells. Therefore, the cellular immunity of vaccine recipients after booster immunization, whether primary or mutant, showed a tendency to increase.

Recently, the state of Kentucky in the United States reported a follow-up survey of people who had recovered from a previous infection. The results showed that among convalescents infected with SARS-CoV-2 from March to December 2020, the risk of reinfection with COVID-19 in unvaccinated people was 2.34 times (95% CI: 1.58–3.47) that in those who completed two doses of the vaccine [24]. We collected peripheral venous blood from recovered patients with COVID-19 (Delta mutant) and detected their primary, Delta-, and Omicron-specific CD8+ T cells. The results showed that SARS-CoV-2-specific CD8+ T cells were increased in convalescents, whereas the opposite was true in vaccinated individuals. At 50 days after booster immunization, the proportion of ancestral-specific CD8+ T cells was higher than that in convalescents, while there was no difference in mutant strains. These results may explain why vaccination offers higher protection than previous COVID-19 infection [25].

## 5. Conclusions

After booster immunization with a high-dose SARS-CoV-2 inactivated vaccine, the ancestral or Delta- and Omicron-specific neutralizing antibodies and cellular immunity all showed an upward trend to varying degrees.

## Figures and Tables

**Figure 1 viruses-14-02016-f001:**
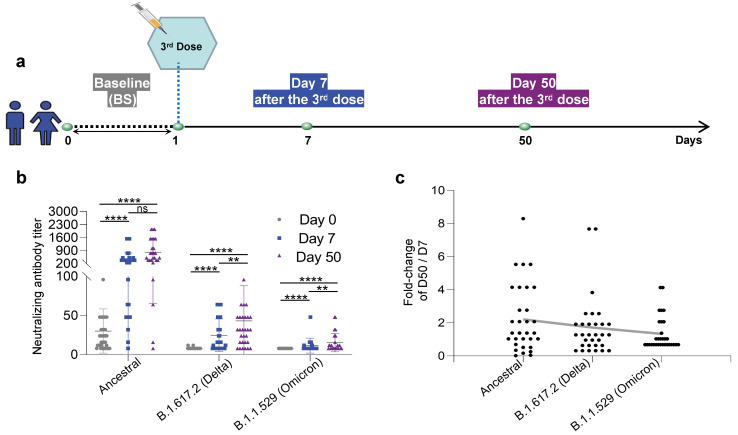
Study design and statistics of anti-SARS-CoV-2 neutralizing antibodies in inactivated SARS-CoV-2 KCONVAC vaccine participants. (**a**) Study design and sample collection timeline. Volunteer participants received 3 doses of the inactivated SARS-CoV-2 vaccine, and blood samples were collected at the indicated time points. (**b**) The inactivated vaccine was titrated with neutralizing antibodies against the ancestral, B.1.617.2 (Delta), and B.1.1.529 (Omicron). The data were presented to all donors. (**c**) Summary statistics of fold-change in neutralizing antibody titers for inactivated vaccines against ancestral, B.1.617.2 (Delta), and B.1.1.529 (Omicron) between 50 and 7 days after the third dose. Data are shown as the mean ± standard deviation (SD). BS: Baseline, prior to the third vaccination dose; D7: 7 days after the third dose; D50: 50 days after the third dose. N = 43 for BS; n = 35 for D7; n = 31 for D50. Each dot represents a single individual. ****: *p* < 0.0001, **: *p* < 0.01, ns: not statistically significant (*p* ≥ 0.05). Consistent *p* value notations are used throughout the paper.

**Figure 2 viruses-14-02016-f002:**
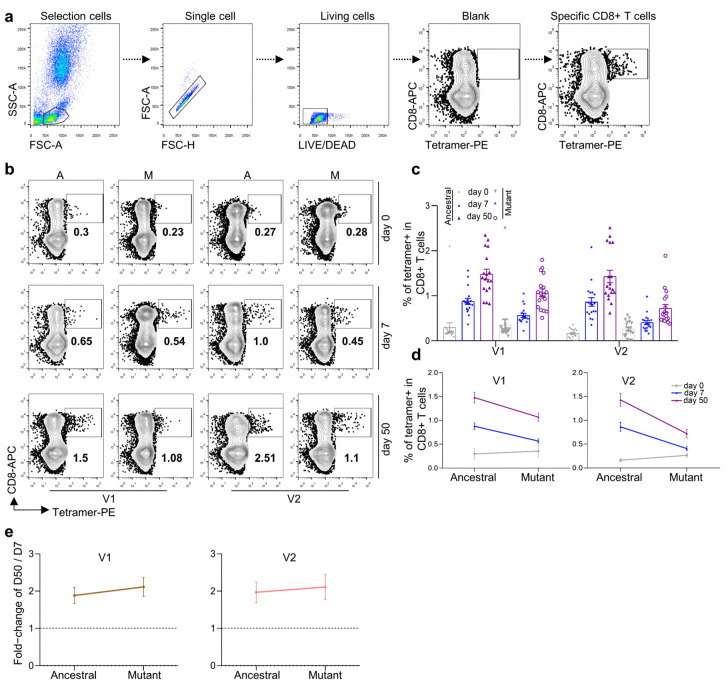
Comparison and characterization of B.1.617.2 (Delta) and B.1.1.529 (Omicron) epitope-specific CD8 T cells between vaccine recipients at 7 and 50 days after the third vaccination dose. (**a**) Flow cytometry gating strategy for SARS-CoV-2 epitope-specific CD8 T cells. (**b**,**c**) Comparison of CD8 T cells specific to ancestral and mutant SARS-CoV-2 epitopes in HLA-A2+ healthy donors 0 (gray), 7 (blue), and 50 (purple) days after the third dose of inactivated SARS-CoV-2 vaccine. Specific CD8 T cells were individually stained with tetramers prepared using ancestral and mutant B.1.617.2 (Delta) and B.1.1.529 (Omicron) epitopes. (**b**) Representative plot of (**c**), n = 18 per group. A: ancestral; M: mutant. Variant strain IDs are the same as listed in Table 1. The flow cytometry gating strategy is shown in (**a**). (**d**) Overall statistics and comparison of CD8 T cells specific to B.1.617.2 (Delta, **left**) and B.1.1.529 (Omicron, **right**) epitopes in HLA-A2+ healthy donors 0 (gray), 7 (blue), and 50 (purple) days after the third dose of inactivated SARS-CoV-2 vaccine. Data shown are the mean ± standard error of the mean (SEM), n = 18 per group. (**e**) Summary statistics of detection fold-change of CD8 T cells specific to B.1.617.2 (Delta, **left**) and B.1.1.529 (Omicron, **right**) epitopes between 50 and 7 days after the third dose. Data shown are the mean ± SD.

**Figure 3 viruses-14-02016-f003:**
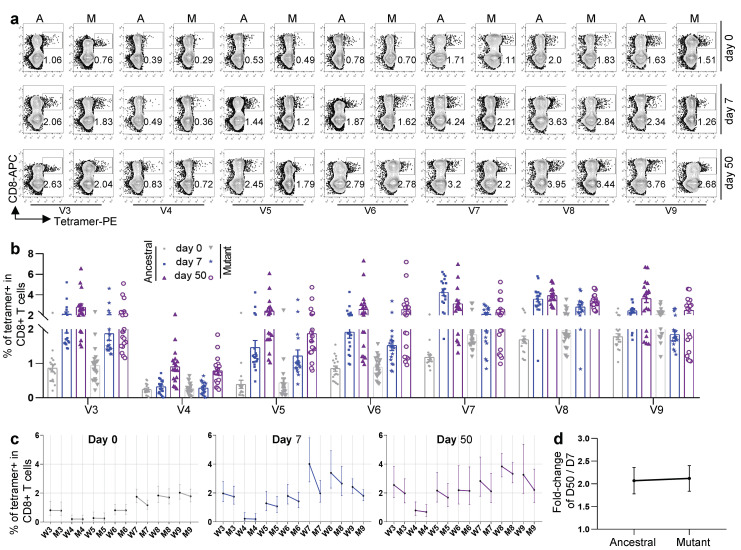
Comparison and characterization of SARS-CoV-2 epitope-specific CD8 T cells between 7 and 50 days after the third dose. (**a**) Representative data for the detection of epitope-specific CD8 T cells in HLA-A2+ healthy donors before and after a third dose of inactivated SARS-CoV-2 vaccine with tetramers prepared using SARS-CoV-2 epitopes. A: ancestral; M: mutant. Variant strain IDs are the same as listed in Table 1. The flow cytometry gating strategy is shown in (**a**). (**b**) Comparison of epitope-specific CD8 T cells between HLA-A2+ healthy young and old donors 0 (gray), 7 (blue), and 50 (purple) days after the third dose of inactivated SARS-CoV-2 vaccine. Specific CD8 T cells were individually stained with tetramers prepared using ancestral and mutant SARS-CoV-2 epitopes. n = 18 per group. (**c**) Comparison of specific CD8 T cells between ancestral and mutant SARS-CoV-2 epitopes in HLA-A2+ donors before the third vaccination with the SARS-CoV-2 inactivated vaccine (**left**), 7 days (**middle**), and 50 days (**right**) after vaccination. Specific CD8 T cells were individually stained with tetramers prepared using ancestral and mutant SARS-CoV-2 epitopes. Variant strain IDs are the same as listed in Table 1. Paired ancestral and mutant epitopes are listed adjacently on the *x*-axis. Data shown are the mean ± SD. (**d**) Summary statistics of the detection fold-change of CD8 T cells specific to SARS-CoV-2 epitopes between 50 and 7 days after the third dose. Data shown are the mean ± SD.

**Figure 4 viruses-14-02016-f004:**
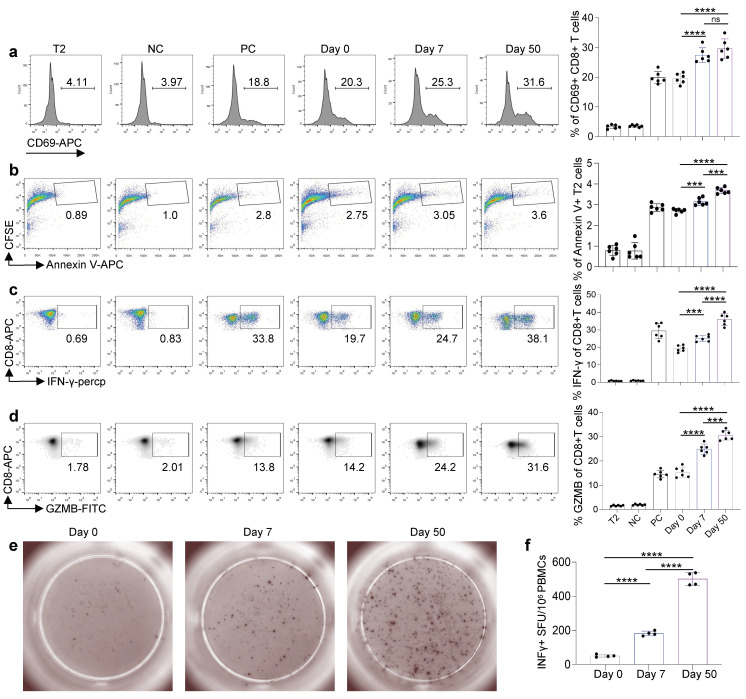
Comparison and characterization of the cytotoxic effects of SARS-CoV-2 epitopes in recipients. (**a**–**d**) Characterization of epitope-specific CD8 T cells after vaccination. CD8 T cells isolated from vaccinated donors after the third dose (days 7 and 50) were cocultivated with T2 cells loaded with SARS-CoV-2 epitopes at a 1:1 ratio and analyzed for the expression of CD69 after 16 h (**a**). The proportion of CFSE+ Annexin V+ T2 cells presenting distinct SARS-CoV-2 antigens after 7 days of culturing with CD8 T cells (**b**). Expression of IFN-γ (**c**) and Granzyme B (**d**) by CD8 T cells after epitope stimulation for 7 days. Day 0: control before stimulation; Day 7: 7 days after the third dose; Day 50: 50 days after the third dose. T2: T2 control cells without any peptide; NC: negative control, T2 cells loaded with EBV virus peptide IVTDFSVIK; PC: positive control, T2 cells loaded with influenza A M1 peptide GILGFVFTL. Data are summarized as the mean ± SD. n = 6 for each group. Statistical significance was determined by one-tailed *t* test or one-way ANOVA. (**e**,**f**) Exemplary microscopy image (**e**) and summary statistics (**f**) for the anti-IFN-γ ELISpot assay on PBMC cells from vaccinated donors (0, 7, and 50 days after the third dose) stimulated by the 9 SARS-CoV-2 ancestral epitopes (mixed; as shown in Table 1). ****: *p* < 0.0001, ***: *p* < 0.001.

**Figure 5 viruses-14-02016-f005:**
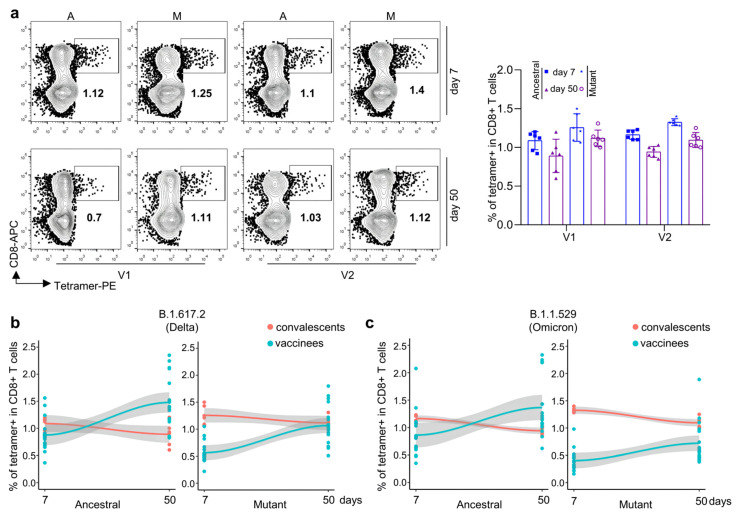
Characterization of B.1.617.2 (Delta) and B.1.1.529 (Omicron) epitope-specific CD8 T cells in convalescent COVID-19 patients. (**a**, **left**) Representative data for the detection of epitope-specific CD8 T cells in HLA-A2+ convalescent COVID-19 patients who recovered from COVID-19 infection at 7 (**top**) and 50 days (**bottom**) with tetramers prepared using B.1.617.2 (Delta) and B.1.1.529 (Omicron) epitopes. (**a**, **right**) Overall statistics and comparison of CD8 T cells specific to B.1.617.2 (Delta) and B.1.1.529 (Omicron) epitopes in convalescent COVID-19 patients. Data are summarized as the mean ± SD. n = 6 for each group. (**b**) Fitted curves for ancestral (**left**) and mutant (**right**) B.1.617.2 (Delta) epitopes at 7 and 50 days after the third dose of vaccination and 7 and 50 days after recovery from COVID-19 infection. (**c**) Fitted curves for ancestral (**left**) and mutant (**right**) B.1.1.529 (Omicron) epitopes at 7 and 50 days after the third dose of vaccination and 7 and 50 days after recovery from COVID-19 infection.

**Figure 6 viruses-14-02016-f006:**
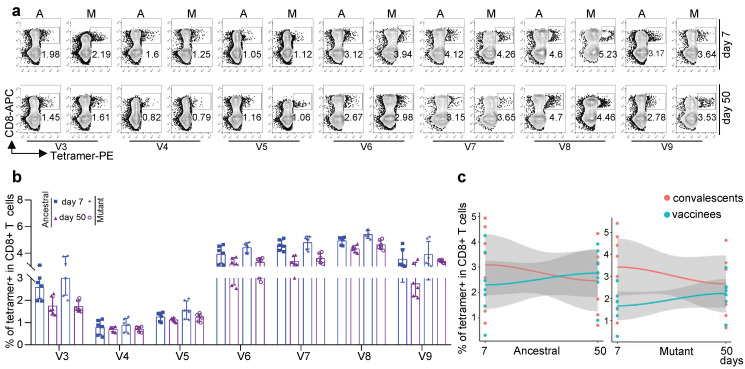
Characterization of mutant epitope-specific CD8 T cells in convalescent COVID-19 patients. (**a**) Representative data for the detection of epitope-specific CD8 T cells in HLA-A2+ convalescent COVID-19 patients who recovered from COVID-19 infection at 7 (**top**) and 50 days (**bottom**) with tetramers prepared using SARS-CoV-2 epitopes. (**b**) Overall statistics and comparison of CD8 T cells specific to ancestral and mutant SARS-CoV-2 epitopes in convalescent COVID-19 patients. Variant strain IDs are the same as listed in Table 1. Data shown are the mean ± SD. n = 6 for each group. (**c**) Fitted curves for ancestral (**left**) and mutant (**right**) epitopes at 7 and 50 days after the third dose of vaccination and 7 and 50 days after recovery from COVID-19 infection.

**Table 1 viruses-14-02016-t001:** Summary of validated epitopes from 9 SARS-CoV-2 variant strains.

ID	Variant Strains	Protein	Ancestral/Mutant	Length	Start Position	End Position	Sequence	Antigenic Value
V1	B.1.617.2 (Delta)	M	Ancestral	9	82	90	IAMACLVGL	1.13
			I82T	9	82	90	**T**AMACLVGL	0.67
V2	B.1.1.529 (Omicron)	S	Ancestral	12	60	71	SNVTWFHAIHVS	0.88
			Δ69-70	10	60	71	SNVTWFHAI**- -**S	0.58
V3	B.1.1.7(Alpha)	ORF1a	Ancestral	10	1707	1716	AANFCALILA	0.44
			A1708D	10	1707	1716	A**D**NFCALILA	0.45
V4	B.1.351 (Beta)	ORF1a	Ancestral	11	3673	3683	SLSGFKLKDCV	0.55
			Δ3675-7	8	3673	3680	SL**- - -**KLKDCV	0.72
V5	P.3	S	Ancestral	9	1171	1179	GINASVVNI	0.69
			V1176F	9	1171	1179	GINAS**F**VNI	1.27
V6	B.1.526.1 (Lota)	ORF8	Ancestral	8	6	13	FLGIITTV	0.65
			T11I	8	6	13	FLGII**I**TV	0.79
V7	B.1.526.2	S	Ancestral	10	2	11	FVFLVLLPLV	0.8
			L5F	10	2	11	FVF**F**VLLPLV	0.8
V8	B.1.618	ORF7b	Ancestral	9	26	34	IIFWFSLEL	0.82
			E33 *	8	26	33	IIFWFSLE-	0.64
V9	B.1.617.3	ORF1a	Ancestral	10	2340	2349	VLGLAAIMQL	0.70
			A2344V	10	2340	2349	VLGL**V**AIMQL	0.83

NOTE: The “*” stands for stop.

## Data Availability

The data presented in this study are available upon request from the corresponding author.

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
