# Peer review of "Effectiveness of Booster Doses of the SARS-CoV-2 Inactivated Vaccine KCONVAC against the Mutant Strains"

_viruses, 2022, doi:10.3390/v14092016_

Round 1

Reviewer 1 Report

The manuscript describes the efficacy of booster doses of the inactivated KCONVAC vaccine against SARS-CoV-2 variants. The Authors showed that both humoral and cellular immunity against the variants was lower than for the Wuhan-Hu strain, but still increased from day 0, through day 7 to day 50 after booster vaccination.

I do not have major comments, the manuscript is well written. However, I see that it was probably prepared for another publisher. Therefore, I have some minor specific comments:

1.     Introduction: the Authors write ”Several studies have demonstrated that the mRNA vaccine can activate humoral immunity against the major SARSCoV-2 mutants that are currently in circulation", but they really only provide two references. Please provide more references, and include in the text what kinds of vaccines (names) have been tested for this.

2.     Material and Methods: when giving the name of the reagent, the name, city, country of the manufacturer should be given in parentheses (in the case of the USA, the state should also be included). This also applies to the vaccine KCONVAC. Do not include catalog numbers. When web references are included, the date of access should be provided.

3.      Results: Table 1 should be in black and white. Alternatively, use red to indicate only mutations.

4.     Discussion: Please state the limitations of the study. Please separate the Conclusions section.

Author Response

Responses to reviewers

Reviewer 1

The manuscript describes the efficacy of booster doses of the inactivated KCONVAC vaccine against SARS-CoV-2 variants. The Authors showed that both humoral and cellular immunity against the variants was lower than for the Wuhan-Hu strain, but still increased from day 0, through day 7 to day 50 after booster vaccination.

I do not have major comments, the manuscript is well written. However, I see that it was probably prepared for another publisher. Therefore, I have some minor specific comments:

Comment 1 - Introduction: the Authors write ”Several studies have demonstrated that the mRNA vaccine can activate humoral immunity against the major SARSCoV-2 mutants that are currently in circulation", but they really only provide two references. Please provide more references, and include in the text what kinds of vaccines (names) have been tested for this.

Response 1 - Thank you for the kind reminder. We have added 2 references of humoral immunity of SARSCoV-2 mutants in the updated manuscript, and attached the vaccine information in detail.

Ref:

  1. Rose, R.; Neumann, F.; Grobe, O.; Lorentz, T.; Fickenscher, H.; Krumbholz, A., Humoral immune response after different SARS-CoV-2 vaccination regimens. BMC medicine 2022, 20, (1), 31.
  2. Chang, X.; Augusto, G.; Liu, X.; Kündig, T.; Vogel, M.; Mohsen, M.; Bachmann, M., BNT162b2 mRNA COVID-19 vaccine induces antibodies of broader cross-reactivity than natural infection, but recognition of mutant viruses is up to 10-fold reduced. Allergy 2021, 76, (9), 2895-2998.

Comment 2 - Material and Methods: when giving the name of the reagent, the name, city, country of the manufacturer should be given in parentheses (in the case of the USA, the state should also be included). This also applies to the vaccine KCONVAC. Do not include catalog numbers. When web references are included, the date of access should be provided.

Response 2 - Thanks for the comment. The vaccine KCONVAC and reagents' information in detail has been supplemented in the updated manuscript which was highlighted in red.

Comment 3 - Results: Table 1 should be in black and white. Alternatively, use red to indicate only mutations.

Response 3 - Thank you for your suggestion. We have revised Table 1 based on your suggestion, see details in Table 1.

Comment 4 - Discussion: Please state the limitations of the study. Please separate the Conclusions section.

Response 4 - Thank you very much for the comment and suggestion. We incorporate the limitations of this study and separate the conclusions section in the updated manuscript, as following:

Due to the limitation of the T2 cell model and the restricted HLA-A2 subtype, this study only focused on HLA-A2-positive populations.  

All together, we appreciate the precious and hard work from the reviewers and editors. We do agree with the general limitation in the initial version. In addition, we did carefully proofreading to minimize grammatical and references errors. The revised above parts have been labeled in red in revised manuscript. Thanks!

Reviewer 2 Report

This is a well prepared manuscript. Here my minor comments for corrections/explanation:
1. Please state in the introduction or materials/methods on why the researcher need to specially select HLA-A2 population.

2. Result section (Page 4, last sentence). "All these results indicated that even lower titers". Please explain what do you mean "even lower titers"? 
3. Figure 1. I think this figure should be increase in size. Using whole page for this figure is better. (the caption and figure should be in one page)

4. Table 1. This table is best to be presented in table format (text, not figure).

5. Figure 3 and Figure 4 are also best to be presented in whole page.

6. Figure 5 caption: There are some virus variant name in small alphabet. There is also wrong spelling of Omicron ( spell as micron).

Author Response

Responses to reviewers

Reviewer: 2

This is a well prepared manuscript. Here my minor comments for corrections / explanation:

Comment 1 - Please state in the introduction or materials/methods on why the researcher need to specially select HLA-A2 population.

Response 1 - Thank you very much for the comment and suggestion. The selected MHC alleles were HLA-A2, the most frequent MHC class I genotype among Chinese population [1,2]. In this study, we performed an HLA-A2 analysis of the donor/patient samples by flow cytometry and pick up the HLA-A2 positive samples for further analysis.

Ref:

1. González-Galarza, F. F.; Takeshita, L. Y. C.; Santos, E. J. M.; Kempson, F.; Maia, M. H. T.; da Silva, A. L. S.; Teles e Silva, A. L.; Ghattaoraya, G. S.; Alfirevic, A.; Jones, A. R.; Middleton, D., Allele frequency net 2015 update: new features for HLA epitopes, KIR and disease and HLA adverse drug reaction associations. Nucleic acids research 2015, 43, (Database issue), D784-D788.

2. He, Y.; Li, J.; Mao, W.; Zhang, D.; Liu, M.; Shan, X.; Zhang, B.; Zhu, C.; Shen, J.; Deng, Z.; Wang, Z.; Yu, W.; Chen, Q.; Guo, W.; Su, P.; Lv, R.; Li, G.; Li, G.; Pei, B.; Jiao, L.; Shen, G.; Liu, Y.; Feng, Z.; Su, Y.; Xie, Y.; Di, W.; Liu, X.; Yang, X.; Wang, J.; Qi, J.; Liu, Q.; Han, Y.; He, J.; Cai, J.; Zhang, Z.; Zhu, F.; Du, D., HLA common and well-documented alleles in China. HLA 2018, 92, (4), 199-205.

Comment 2 - Result section (Page 4, last sentence). "All these results indicated that even lower titers". Please explain what do you mean "even lower titers"?

Response 2 - Thank you very much for the comment. This sentence means that the titration effect of the Delta and Omicron mutant antibodies was weaker than that of the ancestral antibodies.

Comment 3 - Figure 1. I think this figure should be increase in size. Using whole page for this figure is better. (the caption and figure should be in one page)

Response 3 - Thank you for your suggestion. We have revised Figure 1 according to your suggestion.

Comment 4 - Table 1. This table is best to be presented in table format (text, not figure).

Response 4 - Thank you for your suggestion. We have presented table 1 in table format in the updated version.

Comment 5 - Figure 3 and Figure 4 are also best to be presented in whole page.

Response 5 - Thank you for your suggestion. We have presented Figure 3 and Figure 4 in whole page in the updated version.

Comment 6 - Figure 5 caption: There are some virus variant name in small alphabet. There is also wrong spelling of Omicron ( spell as micron).

Response 6 - Thank you for the kind reminder and apologize for the wrong spelling. We revised the virus variant name in the updated version.

All together, we appreciate the precious and hard work from the reviewers and editors. We do agree with the general limitation in the initial version. In addition, we did carefully proofreading to minimize grammatical and references errors. The revised above parts have been labeled in red in revised manuscript. Thanks!